# Increased Emergency Calls during the COVID-19 Pandemic in Saudi Arabia: A National Retrospective Study

**DOI:** 10.3390/healthcare9010014

**Published:** 2020-12-24

**Authors:** Ahmed Al-Wathinani, Attila J. Hertelendy, Sultana Alhurishi, Abdulmajeed Mobrad, Riyadh Alhazmi, Mohammad Altuwaijri, Meshal Alanazi, Raied Alotaibi, Krzysztof Goniewicz

**Affiliations:** 1Department of Emergency Medical Services, Prince Sultan Bin Abdulaziz College Emergency Medical Services, King Saud University, Riyadh 11451, Saudi Arabia; amobrad@ksu.edu.sa (A.M.); rialhazmi@ksu.edu.sa (R.A.); ralotaibi1@KSU.EDU.SA (R.A.); 2Department of Information Systems and Business Analytics, College of Business, Florida International University, Miami, FL 33174, USA; ahertele@fiu.edu; 3Department of Community Health Sciences, College of Applied Medical Sciences, King Saud University, Riyadh 11451, Saudi Arabia; salhurishi@ksu.edu.sa; 4Deputy of General Manager of EMS Administration, Saudi Red Crescent Authority, Riyadh 11451, Saudi Arabia; altuwaijrima@srca.org.sa; 5General Manager of Medical Supply, Saudi Red Crescent Authority, Riyadh 11451, Saudi Arabia; mdalanazi@srca.org.sa; 6Department of Aviation Security, Military University of Aviation, 08521 Dęblin, Poland; k.goniewicz@law.mil.pl

**Keywords:** EMS, Saudi Arabia, call volume, COVID-19, Saudi Red Crescent Authority

## Abstract

The coronavirus 2019 (COVID-19) pandemic has a direct and indirect effect on the different healthcare systems around the world. In this study, we aim to describe the impact on the utilization of emergency medical services (EMS) in Saudi Arabia during the COVID-19 pandemic. We studied cumulative data from emergency calls collected from the SRCA. Data were separated into three periods: before COVID-19 (1 January–29 February 2020), during COVID-19 (1 March–23 April 2020), and during the Holy Month of Ramadan (24 April–23 May 2020). A marked increase of cases was handled during the COVID-19 period compared to the number before pandemic. Increases in all types of cases, except for those related to trauma, occurred during COVID-19, with all regions experiencing increased call volumes during COVID-19 compared with before pandemic. Demand for EMS significantly increased throughout Saudi Arabia during the pandemic period. Use of the mobile application ASAFNY to request an ambulance almost doubled during the pandemic but remained a small fraction of total calls. Altered weekly call patterns and increased call volume during the pandemic indicated not only a need for increased staff but an alteration in staffing patterns.

## 1. Introduction

The coronavirus 2019 (COVID-19) pandemic significantly impacted the healthcare system in Saudi Arabia. The first COVID-19 cases were recorded in Saudi Arabia at the beginning of March 2020 [1,2]. Since the start of the epidemic in March 2020, multiple measures have been implemented to control the disease spread, including societal lockdown, travel and movement restrictions, closure of schools and universities, and the cancelation of mass gatherings and public events [3]. In addition, the government provided ongoing information to the public about the virus and the threat it posed to society in Saudi Arabia. Daily updates about the number of new cases, deaths, and preventive measures taken to reduce transmission were shared with the public. Public health officials and government media campaigns served to educate the public on how to protect themselves and identify symptoms of COVID-19. A strict social distancing policy, including a curfew and complete lockdown, was implemented for 14 days to mitigate the disease spread [4]. These policies may have reduced the number of patient visits to hospitals and emergency departments (EDs), but simultaneously increased the utilization of emergency medical services (EMS).

Almaghlouth et al. reviewed published studies related to COVID-19 studies of medical research from Saudi Arabia, none of which examined the impact of the pandemic on prehospital care [5]. Studies examining the use of prehospital care service in Saudi Arabia during infectious disease outbreaks are limited. Thus, we examined call volume and EMS utilization rates to enable policymakers and operational managers to make informed decisions during the ongoing COVID-19 pandemic and future outbreaks of contagious infectious disease.

The Saudi Red Crescent Authority (SRCA) is the national public EMS organization in Saudi Arabia and provides emergency treatment and transport to an estimated population of 34 million people [6]. EMS can be accessed by either dialing 997 or requesting the service through a mobile app called ASAFNY. This application enables two-way communication between the SRCA dispatch center and the user through the text messaging service SMS (short message service). The user can provide the details of their medical history, including medications they are currently taking, to aid dispatchers in triage and support decision making regarding medical transport.

In a study of how the pandemic affected ED visits in the United States, Hartnett [7] found that ED visits declined between January and May 2020 compared to January and May of 2019. A similar decline was reported for EMS responses in the United States [8]. Understanding how the COVID-19 pandemic affected EMS utilization is important for evaluating health services preparedness and for planning to handle future epidemics to ensure that emergency services are not overwhelmed and that sufficient resources are deployed to meet anticipated demand [9]. This study aims to determine trends in EMS calls during the COVID 19 pandemic across Saudi Arabia, using data recorded by the SRCA.

## 2. Materials and Methods

We reviewed retrospective data from the SRCA pertaining to prehospital transport of emergency cases by paramedic-staffed ambulances. An emergency call is defined as any incident that requires emergency help and involved the dispatch of an ambulance regardless of whether the patient was transported by ambulance to the nearest hospital. The type of emergencies were classified based on the information received by the dispatcher either by phone or through the mobile application ASAFNY. Based on the type of emergency reported by dispatchers, we further classified types into four main groups: medical, trauma, cardiac, and others. This classification is based on the SRCA triage and emergency classification system. A category of “communicable disease” was added within others to represent cases with suspected infectious diseases, including but not limited to COVID-19 cases.

Data for all emergency requests initiated by a client and received by SRCA either by phone or ASAFNY and referred for emergency ambulance dispatch for five months from the 1st of January to the 31st of May 2020 were analyzed. The total number of records retrieved was 378,143. Anonymized data were extracted and include the date of the call, and time of the call (day shift from 06:00 to 18:00 or night shift from 18:00 until 06:00), patient’s location based on the 13-administrative regions of the Kingdom of Saudi Arabia, type of the emergency, and operational information, such as the need for ambulance transportation and further services provided. Age of patients was only available for a subset of calls (during COVID-19 period); therefore, age was not included in the univariate analysis.

The analyses were restricted to EMS requests initiated by individuals. After the initial screening of data, 3733 records were excluded because EMS were provided at a public event staffed by paramedics in attendance and no external activation by 997 or through ASAFNY was initiated. A total of 374,910 fit the inclusion criteria (Figure 1).

Excluded records were those representing EMS provided at a public event with paramedics available and not requiring service initiated by a 997 call or ASAFNY interaction.

To describe the call volume before and during the COVID-19 pandemic, data were divided into calls recorded from the 1st of January until the 29th of February 2020 (before COVID-19) and calls recorded from the 1st of March until the 23rd of April 2020 (during COVID-19). Because the Holy Month of Ramadan occurred from 24th April to 23rd May 2020 and changes in behavior during this period could influence the results, we evaluated data from this month separately.

## 3. Results

### 3.1. Overall Description of Emergency Medical Services Calls

A total of 374,910 emergency calls were received over 5 months from the 1st of January through the 31st of May 2020 (Table 1).

Most calls came by phone (96.91%) rather than by the mobile app. The highest percentage of calls was during April with 27.14% with May showing a similar high volume (24.17%); the lowest call volumes occurred before the pandemic in January and February (~15% each). Call volumes were lowest in the morning (06:00–07:00) and steadily increased until peaking at hour 20 (20:00) (Figure 2).

Consequently, the night shift experienced the highest call volume overall (Table 1). Call volumes were similar throughout the days of the week with the highest percentage of calls on Thursday (14.55%), which corresponds to the beginning of the weekend (Thursday–Saturday) in Saudi Arabia (Table 1). Geographically, the calls were unevenly distributed throughout the Kingdom: 28.77% of calls were from the Makkah Al-Mkarramah region and 22.73% from Al-Riyadh. The Eastern Region had 12.09%. All other regions had less than 10% of the calls.

Age data for most patients were not available [235,839 unknown of 374,910 calls (62.90%)]. For those that were available, 30,903 of 139,071 (22.2%) involved patients 26–35 years old, 25,891 of 139,071 (18.6%) patients 15–25 years old, and 23,412 of 139,071 (16.8%) patients 65 years old and older (Table 1). Most of the calls received were classified as medical (59.94%) with trauma and cardiac representing most of the rest at ~17% each (Table 1). Although calls in the other category represented a small proportion of the total (6.04%), out of these 22,661 other emergency conditions, most (22,199) were calls for suspected communicable disease cases. Slightly more than half (55.68%) of calls resulted in the transport of the patient to the nearest health facility. Refusal of transport by the patient was the most common reason for lack of transport (74.15%) followed by treatment at the scene (15.41%) (Table 1).

### 3.2. Difference in Emergency Medical Services Calls before and during COVID-19

Between 1 March and 23 April 2020 there was a 52.95% change in the number of calls during the COVID-19 pandemic (n = 171,907) compared to the preceding two months (n = 112,394) (Table 2).

Except for calls for trauma, all other types of calls increased. There was a significant increase in calls requested by either phone or the mobile app between the period before and during the COVID-19 period (*p* < 0.05). This increase was greater in requests received through the ASAFNY app during the COVID-19 pandemic, compared to its requests received before the pandemic; however, the app still represented a minor proportion of overall calls (~3% before and during COVID-19).

The percentage of calls was significantly higher for the nightshift with a change of 78.84% during COVID-19. The number of calls increased steeply starting at 14:00 and peaked from 20:00–22:00 before slowly returning to close to, but still above, the number of calls at 07:00 (Figure 3).

Thus, the nightshift experienced prolonged increased call volume throughout the entirety of the shift, not just for a few hours as observed before the pandemic. The pattern of calls over the days of the week was significantly different before and during COVID-19 (*p* < 0.05). Generally, during COVID-19 the peak of calls occurred on Tuesday through Thursday, compared with peaks on Thursday through Saturday (the weekend) before the pandemic (Figure 4), indicating that the pandemic altered weekly staffing patterns. Indeed, rather than having the lowest number of calls as was observed before the pandemic, calls changed by 67.5% on Mondays during the pandemic. Tuesdays had the greatest overall increase in calls with a 69.7% change.

Among all regions, there was a significant increase in the number of calls during COVID-19 (Table 2). The overall pattern of call volume was similar before and during COVID-19 with Makkah Al-Mkarramah and Riyadh having the highest overall numbers of calls (each with >25,000 before COVID-19 and >39,000 during COVID-19). The smallest increases occurred in Makkah Al-Mkarramah with an increase of 9947 calls representing a 27.3% change and Al-Medinah Al-Monawarah with an increase of 2559 calls representing a 22.1% change. In contrast, several areas experienced major increases despite their overall lower number of calls compared with the regions with the highest number of calls: Aljouf calls increased >3-fold during the pandemic compared with pre-pandemic numbers, of the Northern Borders region increased 2.8 times, and Albaha and Jazan both more than doubled in calls.

Not surprisingly, cases classified as others had the greatest change (at 110.18%; although medical cases also had a large change at 84.05% (Table 2). Cardiac cases showed a smaller difference at 26.61%, whereas trauma cases showed a decline of 1641 (a change of −6.11%). Whether some of the medical or cardiac cases related to COVID-19 complications could not be determined from the data collected. Patient refusal of transportation to a hospital was the most common reason for lack of transport before and during COVID-19, although the difference between those transported and not transported was smaller during COVID-19 than before the pandemic (20,179 not transported before COVID-19 and 14,579 not transported during COVID-19).

### 3.3. Emergency Medical Services Calls the Holy Month of Ramadan

An independent subgroup analysis was conducted to compare calls received during the Holy Month of Ramadan with calls received in the other months. There were 88,870 calls made for EMS, which is relatively higher than the monthly average call volume for the whole 5-month period (74,634 calls/month), representing a 16.02% difference of 14,236 calls. Similar to the overall data and the before and during COVID-19 data, the request for EMS were made mainly through phone calls (~96%), and most (61.58%) occurred during the night shift (Table 3).

Unlike the COVID-19 period (Table 2), the distribution of calls during the Holy Month according to the day of the week was more similar to the pre-COVID-19 period with most calls received during the weekend, Thursday (16.91%), Friday (16.58%), and Saturday (13.70%) (Table 3). Geographically, the distribution of calls was similar to the distributions in the overall total and the pre-pandemic and during COVID-19 data with the highest call volumes occurring in Makkah Al-Mkarramah (26.88%), Al-Riyadh (21.95%), and Eastern regions (12.76%). Similar to the pre-COVID-19 call data and the calls from 1 March–23 April, ~60% of EMS requests were medical-related. However, the percentage of the calls classified as suspected COVID-19 was higher at 8%, compared with 6.6% during 1 March–23 April. Similar to the overall percentage of cases transported to hospitals, EMS transported 55.04% of the cases by ambulance during the Holy Month. The most common reason for lack of transport was the refusal by the patient (76.38%), followed by treatment of patients at the scene (15.74%).

## 4. Discussion

Over the three months of the COVID-19 pandemic, a significant increase occurred in call volumes at national and regional levels [10]. The increase in EMS calls contrasts with reported studies that showed a general decrease in emergency utilization in other countries [11]. There was variation in the proportion of increase of calls across regions of Saudi Arabia. A notable increase in calls was observed in areas with lower population density as represented by Al Jouf, Al-Baha, Northern Borders, and Jazan. This larger difference in regions with lower population density was unexpected and suggested that the strict social distancing policy implemented from mid-March to early May was effective in limiting infections in regions with high population density, such as Riyadh and Makkah Al-Mkarramah.

Detailed information about the increase in EMS utilization during the COVID-19 pandemic provides an opportunity to re-evaluate resource allocation and planning efforts. Urgent attention is needed to update emergency preparedness plans that take into consideration the potential for increased call volumes based on geography, time of day, and day of the week. Additionally, different types of emergency situations, such as prolonged pandemics, localized regional epidemics, consequences of natural disasters, or immediate localized threats such as terrorist events [12,13]. An assessment of the EMS systems in the Kingdom is needed to determine if resources are available to provide services based on both standard call volume patterns normally encountered during the year and the capability of the system to respond to multiple crisis events and or prolonged incidents, such as the COVID-19 pandemic, that may strain the EMS system capacity for an undetermined period of time.

The proportion of non-transported cases was 44.32%, which has exceeded previous analyses for similar data during 2014 [1,2,3,4]. The major reason was a patient refusal. The high proportion of patients who refused emergency transportation needs further evaluation. In some cases, effective treatment at the scene can reduce the need for patient transport to an emergency department [14]. Other factors for the high proportion of cases that were not transported could be related to specifics of the case, such as age and gender of the patient and reasons for the call [15]. The SRCA has launched awareness campaigns to increase the utilization of EMS services; future research might focus on evaluating the impact of these educational interventions and trends in EMS usage.

Call demand varied by day of the week and time of day with a significant increase during the night shift, which is similar to findings in other studies. The number of calls for communicable disease reached 12,884 from 1 March to 23 April 2020. This surge in calls likely relates to the following: (i) An increase in suspected COVID-19 cases due to new guidelines for classifying any patient with cough or fever as infectious, and (ii) fear of transporting patients with fever, cough, or other symptoms of COVID-19 in personal vehicles leading to an increase in calls to EMS for an ambulance [16]. Saudi Arabia has a high proportion of EMS calls for trauma due to motor vehicle collisions [17]. Our data showed that the proportion of emergency cases related to trauma markedly decreased during the COVID-19 pandemic, which may be related to the establishment of different levels of curfew and mobility restriction and the occurrence of Ramadan during the period evaluated.

The SRCA launched the ASAFNY app before the pandemic, but the app has not been widely adopted by the public, possibly due to the lack of public awareness concerning this option for contacting EMS and a general lack of awareness about emergency services in Saudi Arabia [18]. The role of telehealthcare has been promoted as a result of COVID-19, which might lead to increased use of ASAFNY in the future. The effectiveness and quality of the EMS dispatch process using the app to activate the system require further study.

## 5. Limitations

The presented analysis has some limitations, as it only includes calls made from January to May 2020. This study includes data from a large cohort of calls (n = 374,910). Generalizing this study to different prehospital care settings in other countries is subject to confounding factors, such as cultural differences and differences in the organizational structure of the EMS system. To date, no similar research has been conducted in Saudi Arabia. The experience gained from this study will form the basis for planned future studies. According to the importance of the problem, it is necessary to conduct further in-depth research in this area.

## 6. Conclusions

The main aim of the study was to investigate the impact on the utilization of emergency medical services (EMS) in Saudi Arabia during the COVID-19 pandemic. During the pandemic period, there was an increase in the overall calls volume, compared with calls volume before the pandemic. Moreover, we observed an alteration in the pattern of received calls. The increase in EMS demand was limited to medical-related emergencies, whereas the demand for trauma-related emergencies decreased in all regions after the national lockdown implementation. The majority of EMS requests were received through a telephone call before and during the pandemic. However, the use of advanced technology applications to request EMS almost doubled during the pandemic period. To our knowledge, this is the only study to date that examines the utilization of the SRCA EMS system at a national level in Saudi Arabia during the COVID-19 pandemic. The study provides valuable insights for policymakers, health systems administrators and planners, and ambulance officials. Furthermore, these results may inform future pandemic preparedness efforts and provide insight regarding the potential effects of lock-down strategies on EMS utilization.

Although much of the attention is focused on diagnosis and treatment of COVID-19, understanding EMS use during COVID-19 provides valuable insight concerning the role of prehospital emergency services at national and regional levels [19,20,21,22,23,24,25,26,27,28,29]. In addition to providing information that can support effective EMS during the current pandemic as infection rates rise and fall, data from this study may help future planning and preparedness initiatives to strategically optimize resources for future pandemics and or events that have the potential strain the capacity of the national EMS system, which could potentially delay or prevent necessary life-saving treatment and transport of the sick and injured to the hospital [30,31]. Responsibility for future preparation must be overseen by decision-makers at the local and national level [32,33]. Multidisciplinary efforts are required to increase the utilization of advanced technology solutions such as the ASAFNY application to request emergency assistance. Given the relative infancy of organized EMS in Saudi Arabia, it is vital to reflect on the quality of EMS preparation [34]. While formal evaluation of training quality is yet to be conducted, it is necessary to increase expenditures on information and education activities that increase public preparation both in the area of threats related to the possible rescue challenges, as well as actions in the EMS system. Additional research should be undertaken to determine the appropriate use of technology, such as the ASAFNY app, to access emergency medical care in Saudi Arabia.

## Figures and Tables

**Figure 1 healthcare-09-00014-f001:**
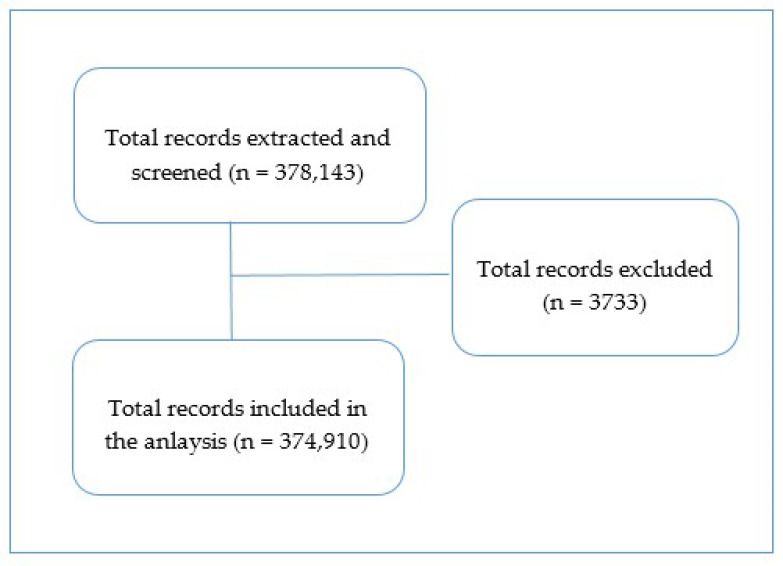
Flowchart of records screening and inclusion.

**Figure 2 healthcare-09-00014-f002:**
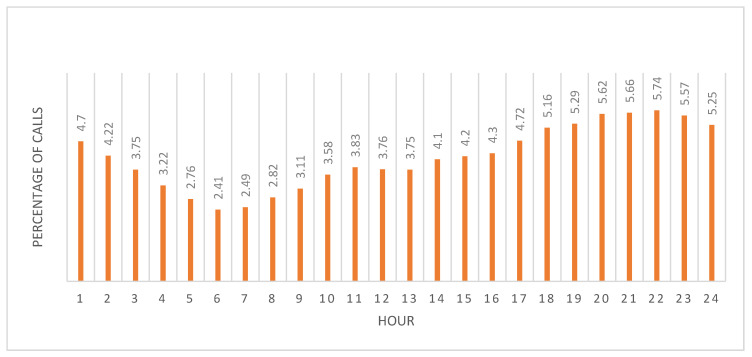
Percentage of emergency medical services (EMS) calls per hour of the day for the entire 5-month period. Day shift corresponded to 06:00 until 18:00; night shift to 18:00 until 06:00.

**Figure 3 healthcare-09-00014-f003:**
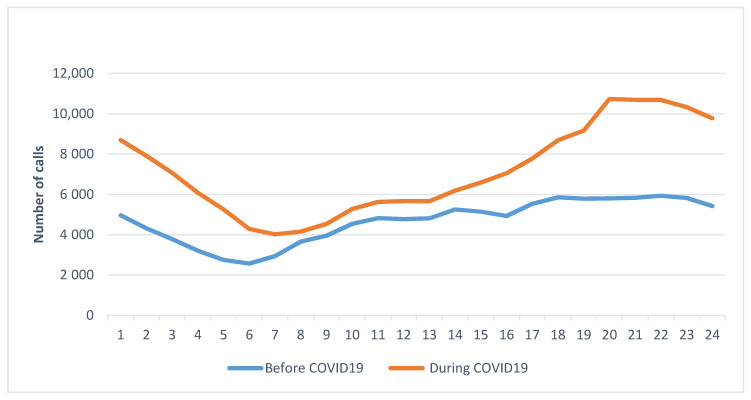
Comparison of the number of EMS calls per hour of the day before and during COVID-19. Before COVID-19 pandemic was defined as 1 January 2020 to 29 February 2020. “During COVID-19” was defined as 1 March 2020 to 23 April 2020.

**Figure 4 healthcare-09-00014-f004:**
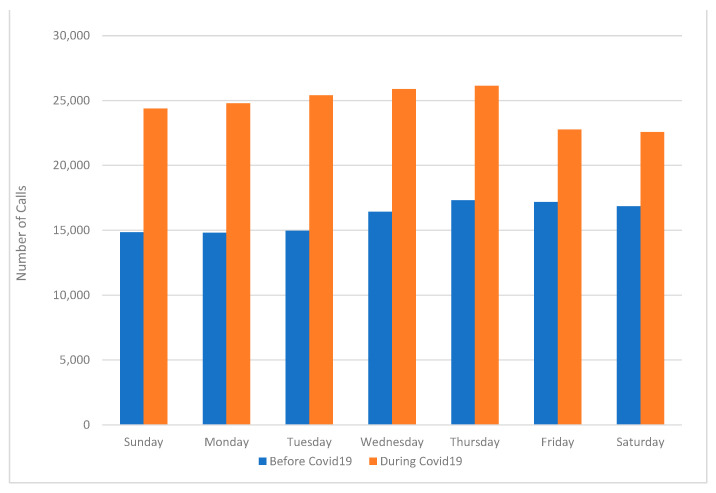
Comparison of the number of EMS calls per day of the week before and during COVID-19. Before COVID-19 pandemic was defined as 1 January 2020 to 29 February 2020. “During COVID-19” was defined as 1 March 2020 to 23 April 2020.

**Table 1 healthcare-09-00014-t001:** Description of Saudi Red Crescent Authority (SRCA) emergency callers (n = 374,910) in Saudi Arabia 1 January 2020 to 31 May 2020.

Variable	Frequency (N)	Percentage (%)
**Age**		
Infant (under 1 year)	1547	1.11
1–14 Years	12,318	8.86
15–25	25,891	18.62
26–35	30,903	22.22
36–45	20,108	14.46
46–55	13,374	9.62
56–65	11,518	8.28
65+	23,412	16.83
Unknown	235,839	62.90
**Month**		
January	57,691	15.39
February	54,703	14.59
March	70,160	18.71
April	101,747	27.14
May	90,609	24.17
**Call per hour-shift**		
Day shift	177,275	47.28
Night shift	197,635	52.72
**Day**		
Sunday	54,133	14.44
Monday	51,940	13.85
Tuesday	52,213	13.93
Wednesday	53,914	14.38
Thursday	54,549	14.55
Friday	54,097	14.43
Saturday	54,064	14.42
**Region**		
Makkah Al-Mkarramah	107,845	28.77
Al-Riyadh	85,208	22.73
Eastern Region	45,313	12.09
Al-Medinah Al-Monawarah	30,018	8.01
Aseer	26,893	7.17
Al-Qassim	17,997	4.80
Jazan	16,747	4.47
Tabouk	10,513	2.80
Hail	8329	2.22
Al-Baha	7549	2.01
Al-Jouf	6976	1.86
Najran	6523	1.74
Northern Borders	4999	1.33
**Type of emergency condition**		
Medical	224,729	59.94
Trauma	64,098	17.10
Cardiac	63,422	16.92
Other	22,661	6.04
**Communicable disease**	22,199	5.92
**Transportation status**		
Transported	208,749	55.68
Non-transported	166,161	44.32
**Reason for non-transported**		
Refuse transportation	123,204	74.15
Treated at scene	25,600	15.41
No injury found	6937	4.17
Death	5989	3.60
No case found	3105	1.87
**Means of EMS request**		
Telephone	363,315	96.91
App (ASAFNY)	11,595	3.09

**Table 2 healthcare-09-00014-t002:** Description of SRCS emergency calls (n = 284,301) in Saudi Arabia before and during the COVID-Pandemic.

	Before COVID-19	During COVID-19	Change in %	*p*
**Variable**				
**Number of EMS calls**	112,394	171,907	52.9%	
**Call per Hour-shift**				<0.05
Day shift	60,525	79,145	+30.8	
Night shift	51,869	92,762	+78.8	
**Day of the week**				<0.05
Sunday	14,853	24,383	+64.2	
Monday	14,800	24,784	+67.5	
Tuesday	14,966	25,397	+69.7	
Wednesday	16,423	25,873	+57.5	
Thursday	17,323	26,146	+50.9	
Friday	17,177	22,754	+32.5	
Saturday	16,852	22,570	+33.9	
**Region**				0.001
Makkah Al-Mkarramah	36,400	46,347	+27.3	
Al-Riyadh	25,672	39,005	+51.9	
Eastern Region	12,066	21,632	+79.3	
Al-Medinah Al-Monawarah	11,602	14,161	+22.1	
Aseer	7048	12,160	+72.5	
Al-Qassim	5059	8215	+62.4	
Jazan	3885	8112	+108.8	
Tabouk	3054	5173	+69.4	
Hail	2309	3921	+69.8	
Al-Baha	1444	3593	+148.8	
Al-Jouf	1194	3661	+206.6	
Najran	1369	2305	+68.4	
Northern Borders	1292	3622	+180.3	
**Type of emergency**				<0.05
Medical	59,011	108,609	+84.0	
Trauma	26,853	25,212	−6.1	
Cardiac	21,149	26,776	+26.6	
Others	5381	11,310	+110.2	
**Communicable disease**	159	12,884	+8003.1	<0.05
**Transportation status**				<0.05
Transported	66,286	93,243	+40.7	
Non-transported	46,108	78,664	+70.6	
**Reason for non-transported**				<0.05
Refuse transportation	31,812	60,023	+88.7	
Treated at scene	7529	11,795	+56.7	
No injury found	2910	2734	−6.0	
Death	2262	2201	−2.7	
No case found	1032	1380	+33.7	
**Means of EMS request**				<0.05
Telephone	110,274	165,650	+50.2	
App (ASAFNY)	2120	6257	+195.1	

“Before COVID-19” was defined as 1 January 2020 to 29 February 2020. “During COVID-19” was defined as 1 March 2020 to 23 April 2020. These data exclude the Holy Month of Ramadan. Significance was considered *p* < 0.05.

**Table 3 healthcare-09-00014-t003:** Description of SRCS emergency calls (n = 88,870) in Saudi Arabia during Ramadan (24 April–23 May 2020).

Variable	Frequency (N)	Percentage
**Call per Hour-shift**		
Day shift	34,144	38.42
Night shift	54,726	61.58
**Day**		
Sunday	11,919	13.41
Monday	11,960	13.46
Tuesday	11,522	12.97
Wednesday	11,529	12.97
Thursday	15,030	16.91
Friday	14,731	16.58
Saturday	15,234	13.70
**Region**		
Makkah Al-Mkarramah	23,888	26.88
Al-Riyadh	19,504	21.95
Eastern Region	11,340	12.76
Al-Medinah Al-Monawarah	6118	6.88
Aseer	7134	8.03
Al-Qassim	4507	5.07
Jazan	4878	5.49
Tabouk	2134	2.40
Hail	2115	2.38
Al-Baha	2308	2.60
Al-Jouf	2165	2.44
Najran	1596	1.80
Northern Borders	1183	1.33
**Type of emergency**		
Medical	55,204	62.12
Trauma	12,216	13.75
Cardiac	15,127	17.02
Others	6323	7.11
**Communicable disease**	7780	8.46
**Transportation status**		
Transported	48,919	55.04
Non-transported	39,951	44.96
**Reason for non-transported**		
Refuse transportation	30,513	76.38
Treated at scene	6285	15.74
No injury found	1243	3.11
Death	1268	3.17
No case found	642	1.60
**Means of EMS request**		
Telephone	85,770	96.51
App (ASAFNY)	3100	3.49

## Data Availability

Datasets used and analysed during the current study are available from the corresponding author on reasonable request.

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
