# Peer review of "Increased Emergency Calls during the COVID-19 Pandemic in Saudi Arabia: A National Retrospective Study"

_healthcare, 2020, doi:10.3390/healthcare9010014_

Round 1
Reviewer 1 Report
Comments and Suggestions for Authors (will be shown to authors)
The manuscript is quite interesting, giving a comprehensive description of the current changes in healthcare needs during the COVID pandemic in Saudi Arabia. It described the increased call volume before and after the pandemic and set aside the period of the Holy Month of Ramadan to avoid possible bias caused by changes in behavior. The results and discussion sections were well written in a logical manner. However, a few minor revisions would be needed to provide additional information.
Abstract
None.
Introduction
None.
Materials and methods
- Page 2, line 77: The major calls were received for the medical issues in this study. Please clarify what the medical group means. How the authors did classify the groups? Please provide criteria for it.
Results
- Page 5, line 120: The authors designed the study with the period before and after the COVID pandemic and the Holy Month of Ramadan. It would be good to show the percentage of EMS calls per hour of the day for each period, not the entire 5-month period in Figure 2.
- Page 7, line 154: Please provide the name of the X-axis as “Hour” in Figure 3.
- Page 7, line 166: I would suggest a bar graph instead of this graph with a linear connecting line.
Discussion
None.
Limitations
None.
Conclusion
None.
Author Response
Response to Reviewer #1
Manuscript ID: healthcare-1038030
Title: Increased Emergency Calls during the COVID-19 Pandemic in Saudi Arabia: A National Retrospective Study
Authors: Ahmed Al-Wathinani, Attila J. Hertelendy, Sultana Alhurishi, Abdulmajeed Mobrad, Riyadh Alhazmi, Mohammad Altuwaijri, Meshal Alanazi, Raied Alotaibi and Krzysztof Goniewicz
Dear Editor,
We would like to express our sincere thanks to you for the positive and constructive comments on our previous manuscript. Based on your comments, we have worked to improve the paper. Our response is provided below, with reviewer comments in italics followed by our response in plain text.
Best regards
Authors
Dear reviewer,
Comment 1: Page 2, line 77: The major calls were received for the medical issues in this study. Please clarify what the medical group means. How the authors did classify the groups? Please provide criteria for it.
Response: Thank you. This classification is based on the Saudi Red Crescent Authority triage and emergency classification system. This information is in the text in the line 74-75
Comment 2 Page 5, line 120: The authors designed the study with the period before and after the COVID pandemic and the Holy Month of Ramadan. It would be good to show the percentage of EMS calls per hour of the day for each period, not the entire 5-month period in Figure 2. Page 7, line 154: Please provide the name of the X-axis as “Hour” in Figure 3.
Response: Thank you. We have improved the quality of figures.
Comment 3: Page 7, line 166: I would suggest a bar graph instead of this graph with a linear connecting line.
Response: Thank you for your comment. We agree and changed graph to bar..
We hope that the manuscript will meet your expectations and that our changes will be satisfactory.
Thank you once again for the valuable tips and time dedicated to review of our work.
Reviewer 2 Report
The issue of the increased utilization of Emergency resources during the COVID-19 Pandemic could be of interest everywhere. The policies adopted all over the world to mitigate the disease spread, as reported by the Authors, are: strict social distancing policy, curfew and/or complete lockdown, ravel and movement restrictions, closure of schools and universities, and the cancelation of mass gatherings and public events. Other Policies have been adopted too (i.e. reducing the access to the hospitals except for emergencies, postponing non-urgent visits, etc.). All these strategies and the additional direct demands of services for COVID-19 complications, have had an impact on healthcare systems and in particular on the management of the Emergency Service. In this specific sector, the analysis of the data on call volume and utilization rates, could offer the opportunity to understand the weaknesses and the strengths of a system. This is a fundamental tool for the decision makers. Unfortunately, contrary to reader’s expectations, the Authors don’t perform this kind of analysis: they report rates without going in deep to understand the reasons of the found differences and without proposing potential solutions to fight the present and, potentially, future similar challenges to the healthcare system.
Author Response
Response to Reviewer #2
Manuscript ID: healthcare-1038030
Title: Increased Emergency Calls during the COVID-19 Pandemic in Saudi Arabia: A National Retrospective Study
Authors: Ahmed Al-Wathinani, Attila J. Hertelendy, Sultana Alhurishi, Abdulmajeed Mobrad, Riyadh Alhazmi, Mohammad Altuwaijri, Meshal Alanazi, Raied Alotaibi and Krzysztof Goniewicz
Dear Editor,
We would like to express our sincere thanks to you for the positive and constructive comments on our previous manuscript. Based on your comments, we have worked to improve the paper. Our response is provided below, with reviewer comments in italics followed by our response in plain text.
Best regards
Krzysztof Goniewicz, Mariusz Goniewicz, Frederick Burkle and Amir Khorram-Manesh
Comment 1: The issue of the increased utilization of Emergency resources during the COVID-19 Pandemic could be of interest everywhere. The policies adopted all over the world to mitigate the disease spread, as reported by the Authors, are: strict social distancing policy, curfew and/or complete lockdown, ravel and movement restrictions, closure of schools and universities, and the cancelation of mass gatherings and public events. Other Policies have been adopted too (i.e. reducing the access to the hospitals except for emergencies, postponing non-urgent visits, etc.). All these strategies and the additional direct demands of services for COVID-19 complications, have had an impact on healthcare systems and in particular on the management of the Emergency Service. In this specific sector, the analysis of the data on call volume and utilization rates, could offer the opportunity to understand the weaknesses and the strengths of a system. This is a fundamental tool for the decision makers. Unfortunately, contrary to reader’s expectations, the Authors don’t perform this kind of analysis: they report rates without going in deep to understand the reasons of the found differences and without proposing potential solutions to fight the present and, potentially, future similar challenges to the healthcare system
Response: Thank you for your comments. We understand you point but cannot agree. We believe our work is solid and important. Based at reviewers comments we tried to improve our work.
Reviewer 3 Report
General remarks
The paper deals with a very important and interesting topic – EMS utilization in Saudi Arabia during the current pandemic. The paper is clearly written and comprehensively referenced to extant scholarship. That said, a few things need tightening though to further enhance the quality of the paper.
Introduction
-The authors should insert a paragraph after line 69 which states how the paper is organised, as this will guide the readers. For instance, ‘The remainder of the paper is organised as follows. The next section outlines our methodology. We subsequently present and discuss our empirical findings and provide concluding remarks’.
Materials and Methods
The following sentence should be moved to the Introduction section: “The first COVID-19 cases were recorded in Saudi Arabia in the beginning of March 2020 [8,9] “ (Line 100). More precisely, it should be placed after the first sentence in the Introduction section.
Discussion
The following paragraph should be moved to the Conclusion section: “To our knowledge this is the only study to date that examines the utilization of the SRCA EMS system at a national level in Saudi Arabia during the COVID-19 pandemic. The study provides valuable insights for policy makers, health systems administrators and planners, and ambulance officials. Furthermore, these results may inform future pandemic preparedness efforts and provide insight regarding the potential effects of lock-down strategies on EMS utilization” (Lines 210-214). The discussion section should focus on interpreting the results.
Conclusions
The conclusion is too brief. The first sentence should restate the aim of the study. Then the authors should proceed to highlight the paper’s core findings. Subsequently, the authors should outline policy implications. These policy recommendations should clearly suggest what policymakers should do in light of the paper’s findings. Put differently, the authors should suggest concrete policy measures that could be adopted by policymakers.
Author Response
Response to Reviewer #3
Manuscript ID: healthcare-1038030
Title: Increased Emergency Calls during the COVID-19 Pandemic in Saudi Arabia: A National Retrospective Study
Authors: Ahmed Al-Wathinani, Attila J. Hertelendy, Sultana Alhurishi, Abdulmajeed Mobrad, Riyadh Alhazmi, Mohammad Altuwaijri, Meshal Alanazi, Raied Alotaibi and Krzysztof Goniewicz
Dear Editor,
We would like to express our sincere thanks to you for the positive and constructive comments on our previous manuscript. Based on your comments, we have worked to improve the paper. Our response is provided below, with reviewer comments in italics followed by our response in plain text.
Best regards
Authors
Dear reviewer,
Comment 1: - The following sentence should be moved to the Introduction section: “The first COVID-19 cases were recorded in Saudi Arabia in the beginning of March 2020 [8,9] “ (Line 100). More precisely, it should be placed after the first sentence in the Introduction section.
Response: Thank you. As suggested we move it into introduction,
Comment 2 The following paragraph should be moved to the Conclusion section: “To our knowledge this is the only study to date that examines the utilization of the SRCA EMS system at a national level in Saudi Arabia during the COVID-19 pandemic. The study provides valuable insights for policy makers, health systems administrators and planners, and ambulance officials. Furthermore, these results may inform future pandemic preparedness efforts and provide insight regarding the potential effects of lock-down strategies on EMS utilization” (Lines 210-214). The discussion section should focus on interpreting the results
Response: Thank you. As suggested we move it into conclusion
Comment 3: The conclusion is too brief. The first sentence should restate the aim of the study. Then the authors should proceed to highlight the paper’s core findings. Subsequently, the authors should outline policy implications. These policy recommendations should clearly suggest what policymakers should do in light of the paper’s findings. Put differently, the authors should suggest concrete policy measures that could be adopted by policymakers.
Response: Good point. Thank you. We have expanded this section. The most important conclusions have been added.
We hope that the manuscript will meet your expectations and that our changes will be satisfactory.
Thank you once again for the valuable tips and time dedicated to review of our work.
Round 2
Reviewer 2 Report
I have not changed my opinion: the Authors reported rates without going in deep to understand the reasons of the found differences and without proposing potential solutions to fight the present and, potentially, future similar challenges to the healthcare system. The additional elements are only minor changes in figures and introduction/conclusion without answering to the comments I did in the previous review.
best regards.
Author Response
I understand the allegations made by the reviewer but we still do not agree. we have added a proper explanation to the conclusions. The work was approved by the other reviewers and, in our opinion, meets the premises of a scientific work